# Standardized Extract from Wastes of Edible Flowers and Snail Mucus Ameliorate Ultraviolet B-Induced Damage in Keratinocytes

**DOI:** 10.3390/ijms241210185

**Published:** 2023-06-15

**Authors:** Luca Vanella, Valeria Consoli, Ilaria Burò, Maria Gulisano, Manuela Stefania Giglio, Ludovica Maugeri, Salvatore Petralia, Angela Castellano, Valeria Sorrenti

**Affiliations:** 1Department of Drug and Health Sciences, University of Catania, 95125 Catania, Italy; valeria_consoli@yahoo.it (V.C.); ilariaburo95@gmail.com (I.B.); maria.gulisano@hotmail.it (M.G.); manuela1189glg@gmail.com (M.S.G.); maugeriludovica@gmail.com (L.M.); salvatore.petralia@unict.it (S.P.); sorrenti@unict.it (V.S.); 2CERNUT–Research Centre for Nutraceuticals and Health Products, University of Catania, 95125 Catania, Italy; 3Mediterranean Nutraceutical Extracts (Medinutrex), Via Vincenzo Giuffrida 202, 95128 Catania, Italy; info@medinutrex.com

**Keywords:** UVB damage, keratinocytes, waste, byproducts, snail mucus, edible flowers, oxidative stress, glutathione

## Abstract

Several studies have highlighted the ability of snail mucus in maintaining healthy skin conditions due to its emollient, regenerative, and protective properties. In particular, mucus derived from *Helix aspersa muller* has already been reported to have beneficial properties such as antimicrobial activity and wound repair capacity. In order to enhance the beneficial effects of snail mucus, a formulation enriched with antioxidant compounds derived from edible flower waste (*Acmella oleracea* L., *Centaurea cyanus* L., *Tagetes erecta* L., *Calendula officinalis* L., and *Moringa oleifera* Lam.) was obtained. UVB damage was used as a model to investigate in vitro the cytoprotective effects of snail mucus and edible flower extract. Results demonstrated that polyphenols from the flower waste extract boosted the antioxidant activity of snail mucus, providing cytoprotective effects in keratinocytes exposed to UVB radiation. Additionally, glutathione content, reactive oxygen species (ROS), and lipid peroxidation levels were reduced following the combined treatment with snail mucus and edible flower waste extract. We demonstrated that flower waste can be considered a valid candidate for cosmeceutical applications due to its potent antioxidant activity. Thus, a new formulation of snail mucus enriched in extracts of edible flower waste could be useful to design innovative and sustainable broadband natural UV-screen cosmeceutical products.

## 1. Introduction

The skin is a protective barrier organ able to maintain the endogenous homeostasis when exposed to environmental agents. It has three main layers: the epidermis, the dermis and the subcutaneous layer. Keratinocytes are the main cells of the epidermis. These cells play a key role in protecting the human body from external insults. Sunlight is the main source of ultraviolet (UV) radiation, among which UVB, UVA, and UVC are included. UVA radiation (with wavelength between 320 and 400 nm) has low energy but is strongly penetrative, generating various effects on the dermis. UVB, characterized by a wavelength between 280 and 320, is the most destructive UV radiation and primarily affects the epidermis of the skin by producing free radicals, reactive oxygen species (ROS), and DNA damage [1,2,3]. When the DNA damage exceeds repair system capacity, normal skin exposed to sunlight may be seriously affected, leading to cell death, photoaging, and carcinogenesis [4,5].

Research has highlighted the ability of snail mucus in maintaining healthy skin conditions due to its emollient, regenerative, and protective properties [6,7].

In particular, mucus derived from *Helix aspersa muller* has already been reported to have beneficial properties such as antimicrobial activity [8,9] and wound repair capacity. Indeed, snail mucus is rich in mucopolysaccharide, which provides good skin hydration grade and enhances adhesion to the skin. Nevertheless, it was reported to be able to stimulate endogenous hyaluronate synthesis, resulting in an increase in viscoelasticity of the skin [10,11]. If present, polyphenols could give added value to snail mucus as they provide additional antioxidant activity useful to counteract ROS production and oxidative damage [7].

More and more studies have recently focused their attention on developing innovative methods for the extraction of bioactive compounds from different kinds of matrices, even from those which are intended as waste or byproducts. From an innovative perspective, in this study, we tried to associate the properties of antioxidant compounds found in edible flowers to the well-known activity of snail mucus to create novel health-promoting products.

Edible flowers are commonly used in human nutrition, and their consumption has increased in recent years. Numerous studies have demonstrated that edible flowers are sources of phenolic compounds with bioactive potential [12,13,14]. However, their increased consumption causes an increase in the waste of edible flowers, as only the first choice of each production can be destined for culinary use, having to comply with rigorous quality and freshness standards. The management of waste and byproducts is a major concern in the world and has an impact on economics and social sectors, as well as on the environment [15]. Therefore, the waste of edible flowers can be valorized with the production of nutraceutical/cosmeceutical extracts. Edible flowers used in the present paper are mainly known for their nutraceutical effects [16,17,18,19]. *Acmella oleracea* L., *Centaurea cyanus* L., *Tagetes erecta* L., *Calendula officinalis* L., and *Moringa oleifera* Lam. were used as they represent the most common species used for food consumption.

The aim of the present study was to evaluate the biological activity of snail mucus and extract of edible flower waste containing bioactive compounds with antioxidant and anti-inflammatory activity for the potential development of a cosmeceutical formulation.

## 2. Results and Discussion

The total phenolic content (TPC) and total carotenoid (TC) content were evaluated spectrophotometrically. The powdered edible flower extract (EFE) was standardized to contain ≥1.00% TPC and ≥0.001% TC. Extracts of edible flower waste (EFE) showed significant antioxidant activity in a concentration-dependent manner, with respect to the scavenger activity of DPPH and inhibition of superoxide anion, as shown in Figure 1A,B. Additionally, the Pearson correlation coefficient was used to directly correlate the antioxidant capacity of the extract to the total polyphenol content (DPPH assay: r = 0.6396, *p* < 0.05; SOD-like activity: r = 0.93, *p* < 0.0005) (Appendix A). On the other hand, low levels of polyphenols justified the poor antioxidant activity of SEM.

The UV/Vis spectra collected on SEM are reported in Figure 1C. In the absorption spectrum of the sample, a band, with a maximum centered at 255 nm was observed. The UV/Vis spectra collected on EFE are reported in Figure 1D. It can be noted that the EFE exhibited maximum absorption at 276 nm, 410 nm, and 560 nm.

The ability of SEM to absorb in the UV/Vis spectrum region, particularly starting below 400 nm, is considered interesting especially concerning cosmetic applications.

The high absorption of SEM might be due to presence of chemical species, such as proteins/peptides and amino acids, absorbing in the region below 400 nm [20].

The UV-induced increase in reactive oxygen species (ROS) and nuclear factor erythroid 2-related factor-2 (Nrf2) inactivation have both been associated with skin-photodamaging processes [21]. Prolonged exposure to UV radiation acts as a stimulus for massive ROS production which may lead to enhanced skin damage as it exceeds the antioxidant system buffering capacity.

As reported by Gubitosa et al. [20], SEM possesses protective activity against sunburns; however, the bioactive compounds present in EFE, thanks to antioxidant properties, might potentiate this effect.

In order to evaluate the potential cytotoxicity of both EFE and SEM, NCTC 2544 cells were treated with different concentrations of the extracts (0.05, 0.1, 0.25, 0.5, 1, 1.5, and 3 mg/mL EFE; 5, 10, 25, 50, 100, 150, and 300 µg/mL SEM) for 24–48 h, and the MTT assay was subsequently performed (Figure 2). No significant effect was observed for both SEM and EFE at 24 h, except for the highest concentration of EFE (3 mg/mL); however, cytotoxicity was observed after 48 h with the highest concentrations of both.

The in vitro photodamage model was established by exposing cells to UVB radiation (λ = 280–300 nm) for 10, 55, 100, and 200 s. As shown in Figure 3A, cell viability was not affected after 10 s exposure; instead, a significant reduction was observed for longer times (23%, 31%, and 38% for 55, 100, and 200 s, respectively).

On the basis of the UV model results, two exposure times (100–200 s) were selected for further experiments. In order to evaluate the protective effect of the extracts on photodamage in terms of cell viability, cells were treated 24 h prior and 24 h post exposure. The rationale on the basis of which a pretreatment was chosen relies on previously reported data by Karunarathne et al. who demonstrated that post-treatment resulted ineffective against UVB-induced cell damage in keratinocytes [22].

Results reported in Figure 3B,D showed the ability of EFE and SEM to counteract 100 s of UV damage at 0.25 mg/mL and 5–10 µg/mL, respectively, leading to an increased survival rate. However, the effects were not remarkable after 200 s of exposure (Figure 3C,E) probably due to irreversible damage caused by prolonged irradiation.

Preliminary cytotoxicity tests were performed to evaluate whether extract cotreatment could result in reduced cell viability; thus, 0.25–0.5 mg/mL EFE and 5–10 µg/mL SEM were used for combination treatments. No significant toxicity was observed after combination treatment (Appendix A).

The selected combinations were then used as previously described for an evaluation of the protective effect in comparison to the UV model damage.

Combination treatment was able to weakly reduce UV damage in terms of cell viability (Appendix A); however, we observed significant antioxidant activity in vitro, as reported in Figure 4. The oxidative stress condition was assessed by measuring ROS levels (Figure 4A), RSH cellular content (Figure 4B), and lipid peroxidation (Figure 4C), showing a synergistic effect of the cotreatment in addition to lipid peroxide (LOOH) levels.

In particular, cotreatment was able to decrease ROS and LOOH, as well as consistently increase glutathione levels. As a fundamental intracellular antioxidant defense mechanism, glutathione is essential for the maintenance of redox homeostasis; thus, restoration of its reduced form and its ex novo synthesis can counteract UV-induced oxidative stress [23].

In order to evaluate the intrinsic antioxidant activity of EFE and support the correlation with the high polyphenol content, antioxidant gene expression was measured. Results showed that heme oxygenase 1 (HO-1), glutamate–cysteine ligase (GCLC), and glutathione reductase (GSR) levels were significantly increased after EFE treatment in a dose-dependent manner (Appendix A), highlighting the antioxidant activity of the extract in vitro.

## 3. Materials and Methods

### 3.1. EFE Composition

The powdered edible flower extract (EFE) (batch no. 1/22) employed in this study was produced by Medinutrex (Catania, Italy). Edible flowers used in the present study consisted of *Acmella oleracea* L. flower (20%), *Centaurea cyanus* L. flower (20%), *Calendula officinalis* L. flower (20%), *Tagetes erecta* L. flower (20%), and *Moringa oleifera* Lam. flower (20%). Briefly, the extract was prepared from dried and ground edible flowers mixed with hydroalcoholic solutions (food grade) and then filtered. The filtrate was concentrated and then spray-dryed to obtain the standardized extract.

### 3.2. Determination of Total Polyphenol Content of EFE

To determine the overall amount of polyphenols in EFE, the Folin–Ciocâlteu colorimetric method was employed [24]. The findings were then reported in grams of gallic acid equivalents (GAE) per 100 g of EFE, which is a commonly used measure of total phenolic content.

### 3.3. Determination of Total Carotenoid Content of EFE

Five grams of EFE was mixed with 50 mL of an extraction solvent (hexane/acetone/ethanol, 50:25:25, *v*/*v*) for 20 min. The organic phase, which contained the carotenoids, was separated and used for analysis after appropriate dilution with hexane. Total carotenoid content was determined by measuring the absorbance at 450 nm of an aliquot of the hexane extract. The results were then reported in grams of β-carotene per 100 g of EFE.

### 3.4. Helix Mucus Collection and Composition Evaluation

The Helix aspersa snails were fostered in the private snail farm “La lumaca Madonita” (Campofelice di Roccella 90010—Palermo, Italy, https://www.lumacamadonita.it/, accessed on 3 March 2023). Helix aspersa mucus was collected by La Lumaca Madonita (Campofelice di Roccella, Palermo, Italy). Snail mucus extraction was performed according to a cruelty-free system using the patented extractor machine EXTRACTA (https://www.lumacamadonita.it/bava-di-lumaca/, accessed on 3 March 2023). The machine, designed to minimize the suffering of the animals, manages to give a very pure mucus extracted without the use of water or irritating chemical agents. Snail mucus was standardized, characterized, and then microfiltered with potassium sorbate and sodium benzoate, before storing at 4 °C or −80 °C. For in vitro experiments, Helix mucus was diluted in culture medium, and its concentration was expressed as mg proteins/mL.

### 3.5. DPPH Assay

The DPPH radical-scavenging capacity of the edible flower extract (EFE) at different concentrations (0.05, 0.1, 0.25, 0.5, 1, 1.5, 3, 6, and 10 mg/mL) was evaluated as previously reported [25].

### 3.6. SOD-Like Activity

SOD-like activity was measured using the pyrogallol autoxidation method as described in [26]. The percentage inhibition of pyrogallol autoxidation was calculated as follows: % inhibition of pyrogallol autoxidation = [1 − (ΔA/ΔAmax)] × 100, where ΔA is the absorbance change due to pyrogallol autoxidation in the sample reaction system, and ΔAmax is the absorbance change due to pyrogallol autoxidation in the control (w/o samples).

One unit of SOD activity represents the amount required for pyrogallol autoxidation 50% inhibition/min.

### 3.7. UV Spectra

The UV/visible absorption spectra of snail mucus (5 μg/µL) and EFE (0.25 μg/mL) were collected using a spectrophotometer (Perkin Elmer 365, Perkin Elmer, Waltham, MA, USA) in the range of 200–600 nm, at a 1 nm·s^−1^ scan rate. The standard quartz cuvette length of 1 mm was used.

### 3.8. Cell Culture and Viability Assay

The human keratinocyte cell line NCTC 2544 (Interlab Cell Line Collection, Genoa, Italy) was cultured in Eagle’s minimum essential medium (EMEM) with 4.5 g/L glucose, supplemented with 10% FBS and 1% pen/strep solution, and maintained at 37 °C and 5% CO_2_. NCTC 2544 cells were seeded at a concentration of 1 × 10^4^ cells per well of a 96-well plate. The cultures were maintained in the absence or presence of the different concentrations of the extracts (EFE 0.05, 0.1, 0.25, 0.5, 1, 1.5, and 3 mg/mL; SEM 5, 10, 25, 50, 100, 150, and 300 µg/mL) for 24 and 48 h. Cell viability was determined using the MTT assay as previously described [27]. Absorbance was detected at λ = 570 nm in a microplate reader (Biotek Synergy-HT, Winooski, VT, USA). Eight replicate wells were used for each group, and three separate experiments were performed.

### 3.9. Cell Treatments and UV Model Establishment

In order to establish a UVB model, a photoreactor was used with reflecting wall supplied with a lamp with emission centered at λ = 280–300 nm (UVB) (~2 mW/cm^2^). Irradiation doses were calculated using the following formula:Irradiation dose (mJ/cm^2^) = exposure time (s) × irradiance (mW/cm^2^).

Cells were seeded at a concentration of 2.5 × 10^5^ per well in a six-well plate and maintained at 37 °C and 5% CO_2_. After 24 h, cells were exposed to UVB radiation for 10, 55, 100, and 200 s corresponding to 20, 110, 200, and 400 mJ/cm^2^ respectively.

Cell viability was assessed for the UVB model, and 200–400 mJ/cm^2^ irradiation doses were selected for the subsequent experiments. Treatments with the extracts, as reported above, were performed 24 h prior and 24 h post exposure.

### 3.10. Measurement of ROS Levels

Reactive oxygen species (ROS) levels were evaluated using fluorescent probe DCFH-DA as previously described [28]. Results were expressed as the fluorescence intensity (AU)/proteins (mg/mL).

### 3.11. Thiol (RSH) Group Determination

The measurement of thiol group (RSH) concentration was performed as it reflects almost 90% of GSH cellular content. The RSH content was measured as previously described [29]. The experiment was performed at least three times, and the results were expressed as pmol/µL.

### 3.12. Measurement of Lipid Peroxidation

Lipid peroxidation was evaluated using the xylenol orange-based method as previously described [29]. The experiments were conducted at least three times, and the results were expressed as a percentage of control.

### 3.13. RNA Extraction and Quantitative Real-Time PCR Analysis

Cells were treated with EFE at different concentrations (0.05, 0.1, 0.25, 0.5, 1, and 1.5 mg/mL) for 24 h and then harvested for RNA extraction using the Trizol reagent (Invitrogen, Carlsbad, CA, USA). The Applied Biosystem (Foster City, CA, USA) reverse transcription reagent was used to obtain first-strand cDNA, and then qRT-PCR analysis was performed in a Step One Fast Real-Time PCR System Applied Biosystems using the SYBR Green PCR Master Mix (Life Technologies, Monza MB, Italy) to evaluate antioxidant gene expression (HO-1, GSR, and GCLC). Results were normalized with the housekeeping gene GAPDH using a comparative 2^−ΔΔCt^ method.

### 3.14. Statistical Analysis

Each analysis represented the results of at least three independent experiments. Fisher’s method was used to obtain statistical significance (*p* < 0.05) of the differences among the experimental groups. For a comparison of treatment groups, the null hypothesis was tested by either a single-factor analysis of variance (ANOVA) for multiple groups or an unpaired *t*-test for two groups, and the data were presented as the means ± SEM. The correlation analysis results of antioxidant activity with total polyphenols content were expressed as Pearson correlation coefficients.

## 4. Conclusions

Snail mucus from *Helix aspersa muller* has been reported to have several therapeutic properties. Nevertheless, its beneficial effects can be enhanced. Although their consumption has increased in recent years, edible flowers are still considered part of a niche market, suffering from some limitations such as high cost of production due to the high quality standards required, which impose a high level of selection, leading to a high discard rate. We demonstrated that the waste of these varieties of flowers can be considered a good candidate for cosmetic applications, due to the high content of antioxidants. Thus, a new formulation of snail mucus enriched in extracts of edible flower (*Acmella oleracea* L., *Centaurea cyanus* L., *Tagetes erecta* L., *Calendula officinalis* L., and *Moringa oleifera* Lam.) waste containing bioactive compounds with antioxidant activity could be useful to design broadband natural UV-screen cosmeceutical products.

## Figures and Tables

**Figure 1 ijms-24-10185-f001:**
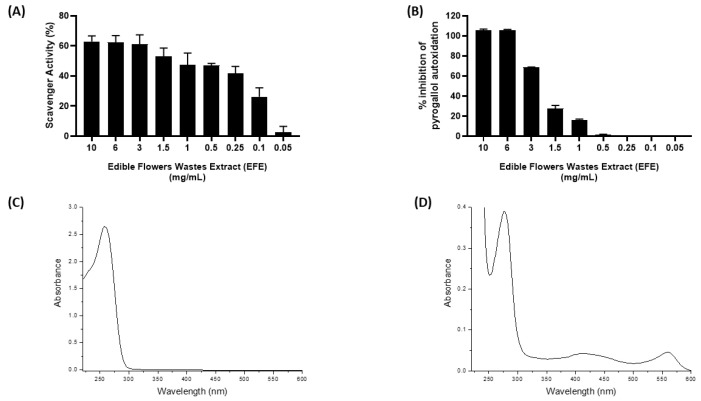
Evaluation of EFE antioxidant activity in a cell-free method (**A**,**B**). Optical absorption spectra (l = 1 mm) for acqueous solutions of SEM (5 µg·µL^−1^) (**C**) and EFE (0.25 µg·mL^−1^) (**D**).

**Figure 2 ijms-24-10185-f002:**
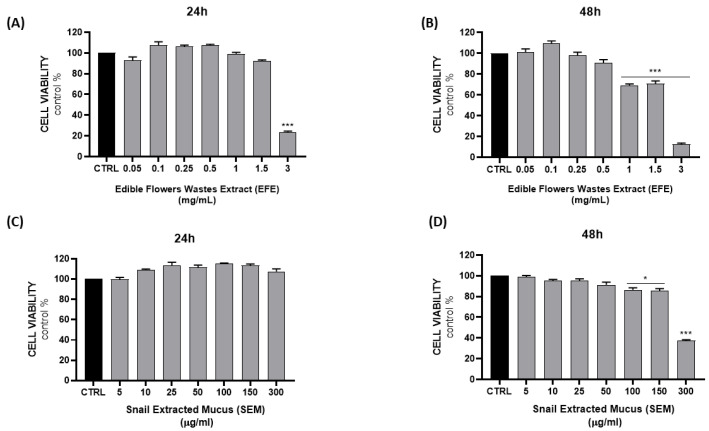
Evaluation of cell viability after 24–48 h of treatment with different concentrations of EFE (**A**,**B**) and SEM (**C**,**D**). Results are expressed as the mean ± SEM. (* *p* < 0.05; *** *p* < 0.0005 vs. CTRL).

**Figure 3 ijms-24-10185-f003:**
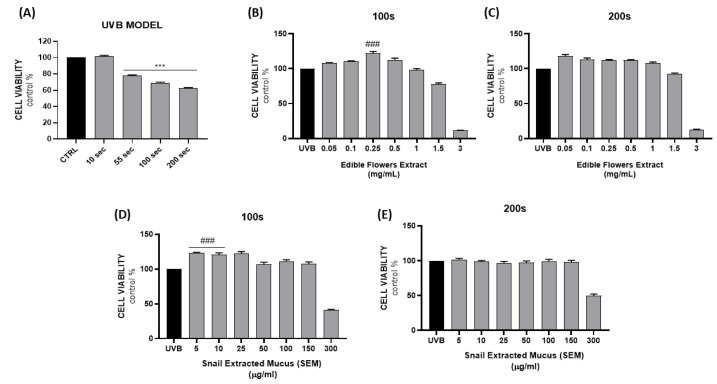
Assessment of cell viability following UVB model establishment (**A**). Evaluation of EFE and SEM protective effects on NCTC 2544 cells after UVB exposure (100–200 s) (**B**–**E**). Results are expressed as the mean ± SEM. (*** *p* < 0.0005 vs. CTRL; ### *p* < 0.0005 vs. UVB).

**Figure 4 ijms-24-10185-f004:**
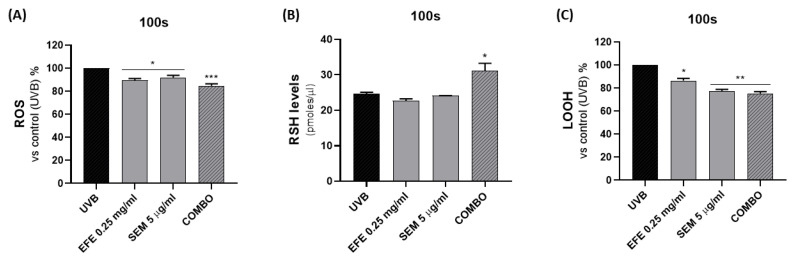
In vitro evaluation of EFE and SEM antioxidant activity alone and in combination on NCTC 2544 cells after UVB exposure (100 s). Reactive oxygens species (**A**), RSH (**B**) and LOOH (**C**) levels were evaluated (COMBO: SEM 5 µg/mL + EFE 0.25 mg/mL)**.** Results are expressed as the mean ± SEM. (* *p* < 0.05; ** *p* < 0.005; *** *p* < 0.0005 vs. CTRL).

## Data Availability

Not applicable.

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
