# Peer review of "Standardized Extract from Wastes of Edible Flowers and Snail Mucus Ameliorate Ultraviolet B-Induced Damage in Keratinocytes"

_ijms, 2023, doi:10.3390/ijms241210185_

Round 1
Reviewer 1 Report
This manuscript presents a study on the evaluation of oxidative stress status through the assessment of ROS levels, RSH cellular content, and lipid peroxidation. This topic is meaningful. But there are still some things that need to be revised carefully.
1. Add short conclusions after each result section in the previously provided article. This will help the readers to better understand the flow of the article and the implications of the results.
2. In Figure 4C of the manuscript, it appears that the combination treatment does not demonstrate a synergistic effect, as there is no significant difference observed compared to the SEM group.
3. In the figure legend, it is important to clearly specify what "COMBO" refers to, as this abbreviation is not immediately understandable to readers.
4. In the study, the authors evaluated oxidative stress conditions by measuring ROS levels, RSH cellular content, and lipid peroxidation. However, it would be beneficial if the authors also considered analyzing gene expression related to oxidative stress. Assessing the expression levels of oxidative stress-related genes could provide additional insights into the molecular mechanisms underlying the observed changes in ROS, RSH, and lipid peroxidation.
Overall, with the recommended revisions, including the incorporation of gene expression analysis and addressing the observed limitations, this manuscript has the potential to make valuable contributions to the field of oxidative stress research. I recommend accepting the manuscript for publication after the authors have addressed the mentioned revisions.
Author Response
We appreciate the reviewers’ comments and we have improved the manuscript as suggested. All changes have been highlighted in yellow.
REVIEWER 1
This manuscript presents a study on the evaluation of oxidative stress status through the assessment of ROS levels, RSH cellular content, and lipid peroxidation. This topic is meaningful. But there are still some things that need to be revised carefully.
- Add short conclusions after each result section in the previously provided article. This will help the readers to better understand the flow of the article and the implications of the results.
Manuscript has been improved as suggested.
- In Figure 4C of the manuscript, it appears that the combination treatment does not demonstrate a synergistic effect, as there is no significant difference observed compared to the SEM group.
We thank you for the comment, we have addressed the lack of synergism for LOOH levels.
- In the figure legend, it is important to clearly specify what "COMBO" refers to, as this abbreviation is not immediately understandable to readers.
COMBO treatment has been specified in figure legends.
- In the study, the authors evaluated oxidative stress conditions by measuring ROS levels, RSH cellular content, and lipid peroxidation. However, it would be beneficial if the authors also considered analyzing gene expression related to oxidative stress. Assessing the expression levels of oxidative stress-related genes could provide additional insights into the molecular mechanisms underlying the observed changes in ROS, RSH, and lipid peroxidation.
Levels of oxidative stress related genes have been evaluated, in particular results showed that heme oxygenase 1 (HO-1), glutamate-cysteine ligase (GCLC) and glutathione reductase (GSR) levels were significantly increased after EFE treatment in a dose-dependent manner (figure S2).
Overall, with the recommended revisions, including the incorporation of gene expression analysis and addressing the observed limitations, this manuscript has the potential to make valuable contributions to the field of oxidative stress research. I recommend accepting the manuscript for publication after the authors have addressed the mentioned revisions.
REVIEWER 2
This paper reports the Standardized Extract from Wastes of Edible Flowers and Snail Mucus Ameliorate Ultraviolet B-induced Damage in Keratinocytes. The manuscript is well prepared and the subject is interesting. In my opinion, the manuscript is in a position to be accepted for publication after major revision. Here are some comments on the manuscript:
- For the first citation, the binomial name of all studied species should be followed by the authority.
It has been modified as suggested.
- In the abstract section, the authors use many abbreviations like GSH content, ROS and LOOH which make this part difficult to understand. The authors should remove some abbreviations.
Abstract has been corrected.
- The discussion of the obtained resullts is very poor. The authors must compare their results with other works.
Additional references have been added to support our results.
- In the material section, the authors should give more information about the studied edible flowers wastes and the studied formulation.
We modified the main text since these preliminary results should eventually support the development of a potential new formulation, which has not been formulated yet.
- In conclusion part, the authors must remove the references (19-22).
We removed references from the conclusion section.
- Check the whole paper to remove spelling mistakes and insert blanks where necessary.
The manuscript has been revised.
- All the references should be cited in the text according to “instructions for authors” of the journal. In the same way, the references list should be revised and adapted to “instruction for authors” of the journal.
Reference list has been revised.
REVIEWER 3
I have read the communication ‘‘Standardized Extract from Wastes of Edible Flowers and Snail Mucus Ameliorate Ultraviolet B-induced Damage in Keratinocytes’’ by Vanella et al. I must say that this study is preliminary, with no specific focus nor clear objectives. The results presented are not in any way meaningfully discussed. The individual plant antioxidant potential has not been assessed. Because Pearson’s correlation was not done for the relationship between the observed antioxidant activity, and phenolic content, it is not useful to assume that the observed antioxidant activity could be due to the phenolics.
We adopted the Pearson correlation coefficient (PCC), also referred to as Pearson’s r, to express the strength and direction of the linear relationship of correlation between antioxidant activity (as both DPPH scavenger activity and SOD-like activity) and total polyphenols content of the edible flowers’ extract.
- Abstract
Nowadays, several studies >> research
Text has been modified.
-Cytotoxicity results not hinted on
Results discussion has been implemented.
All abbreviations should be expanded at first use. Names of animals and plants should be written following the correct binomial nomenclature e.g.
It has been corrected as suggested.
- H. aspersa muller >> Helix aspersa muller. Please check to reconfirm if this species is now not classified as Cornu aspersum (O. F. Müller, 1774).
- Centaurea Cianus >> Centaurea cyanus L.
- Moringa Oleifera >> and Moringa oleifera
Text has been modified as suggested.
- Introduction
It is not argued out why the flowers of these species were selected.
Acmella oleracea L., Centaurea cyanus L., Tagetes erecta L., Calendula officinalis L. and Moringa oleifera Lam. were used in this study as they represent the most common species used for food consumption and thus they constitute the discarded products of the elite food industry.
- Results and discussions
In Figure 1, I would present only (B) because it is the most comparable. These are essentially the same thing.
Data reported in figure 1 A and B represent two different antioxidant mechanisms of action of the extract, as scavenger of DPPH radical and inhibitor of pyrogallol autoxidation mimicking superoxide dismutase activity. Indeed, results obtained by the assays show different behaviors.
-Avoid describing the methodology under this section. If not, consider giving methodology first before RESULTS.
It has been done.
-There is certainly no discussion that can be seen under this section.
As a communication article the sections “Results” and “Discussions” were reported together.
However, discussion has been improved as suggested.
- General comment
-There are overlaps seen in a reasonable percentage of the manuscript (see attached report)
The manuscript has been modified as suggested.
-Sources of the methods used could be preferably cited.
Citations were added as suggested.
Reviewer 2 Report
This paper reports the Standardized Extract from Wastes of Edible Flowers and Snail Mucus Ameliorate Ultraviolet B-induced Damage in Keratinocytes. The manuscript is well prepared and the subject is interesting. In my opinion, the manuscript is in a position to be accepted for publication after major revision. Here are some comments on the manuscript:
- For the first citation, the binomial name of all studied species should be followed by the authority.
- In the abstract section, the authors use many abbreviations like GSH content, ROS and LOOH which make this part difficult to understand. The authors should remove some abbreviations.
- The discussion of the obtained resullts is very poor. The authors must compare their results with other works.
- In the material section, the authors should give more information about the studied edible flowers wastes and the studied formulation.
- In conclusion part, the authors must remove the references (19-22).
- Check the whole paper to remove spelling mistakes and insert blanks where necessary.
- All the references should be cited in the text according to “instructions for authors” of the journal. In the same way, the references list should be revised and adapted to “instruction for authors” of the journal.
Author Response

(The authors gave the same response as above.)

Reviewer 3 Report
I have read the communication ‘‘Standardized Extract from Wastes of Edible Flowers and Snail Mucus Ameliorate Ultraviolet B-induced Damage in Keratinocytes’’ by Vanella et al. I must say that this study is preliminary, with no specific focus nor clear objectives. The results presented are not in any way meaningfully discussed. The individual plant antioxidant potential has not been assessed. Because Pearson’s correlation was not done for the relationship between the observed antioxidant activity, and phenolic content, it is not useful to assume that the observed antioxidant activity could be due to the phenolics.
1. Abstract
Nowadays, several studies >> research
-Cytotoxicity results not hinted on
All abbreviations should be expanded at first use. Names of animals and plants should be written following the correct binomial nomenclature e.g.,
- H. aspersa muller >> Helix aspersa muller. Please check to reconfirm if this species is now not classified as Cornu aspersum (O. F. Müller, 1774).
- Centaurea Cianus >> Centaurea cyanus L.
- Moringa Oleifera >> and Moringa oleifera
2. Introduction
It is not argued out why the flowers of these species were selected.
3. Results and discussions
In Figure 1, I would present only (B) because it is the most comparable. These are essentially the same thing.
-Avoid describing the methodology under this section. If not, consider giving methodology first before RESULTS.
-There is certainly no discussion that can be seen under this section.
4. General comment
-There are overlaps seen in a reasonable percentage of the manuscript (see attached report)
-Sources of the methods used could be preferably cited.

Should be improved
Author Response

(The authors gave the same response as above.)

Round 2
Reviewer 2 Report
Acceot
Author Response
We thank the reviewer for the valuable comments.
Reviewer 3 Report
The authors have revised the manuscript agreeably, and addressed most of my concerns. But there are some minor corrections that need to be done as follows;
-For plant names, please take care that only the genus and species names are italicized, excluding authority names.
- I don’t seem to find any information regarding the TPC and carotenoids content of the extracts in the results section though they are determined as per METHODS.
-In L128 and L129, what does LOOH mean? Please expand this at first use.
-To the best of my knowledge, the correlation coefficients (r = 0.6396) in Figure S3 indicates that the phenolics actually does not contribute to the observed antioxidant and SOD activities. Ideally, if the correlation were negative, this would imply that the total phenolic compounds play a significant role in increasing DPPH radical scavenging activity by the extracts (see e.g., Kim et al., 2020; Omara et al., 2021; Zengin et al., 2019).
Nakiguli et al. (2022). Phytochemical Composition and Antioxidant Activities of Phosphate Buffered Saline and Aqueous Extracts of Aloe barbadensis Miller Leaf Latex and Gel from Three Counties of Kenya. Asian Journal of Applied Chemistry Research, 17-32.
Kim JS, Lee JH. Correlation between Solid Content and Antioxidant Activities in Umbelliferae Salad Plants. Prevent Nutr Food Sci. 2020; 25:84–92.
Zengin G, Mahomoodally MF, Paksoy MY, Picot-Allain C, Glamocilja J, Sokovic M, et al. Phytochemical characterization and bioactivities of five Apiaceae species: natural sources for novel ingredients. Ind Crop Products. 2019; 135:107–121.
Omara et al. (2021) Intraspecific Variation of Phytochemicals, Antioxidant, and Antibacterial Activities of Different Solvent Extracts of Albizia coriaria Leaves from Some Agroecological Zones of Uganda", Evidence-Based Complementary and Alternative Medicine, 2335454, 14 pages, 2021. https://doi.org/10.1155/2021/2335454
L141: acqueous >> aqueous
Grammar needs to be checked throughout
Author Response
We thank the reviewer for the comments. We improved the manuscript following your suggestions. As concern Pearson Correlation coefficient, we plotted TPC with percentage of DPPH inhibition and SOD-like activity, thus it is a direct correlation (r=positive value). On the contrary expressing antioxidant activity as IC50 value may result as a negative correlation indicating also in this case an elevated antioxidant activity.